# Generalization of Heterogeneous Multi-Robot Policies via Awareness and Communication of Capabilities[†]

**Pierce Howell**[1*]**, Max Rudolph**[2*]**, Reza Torbati**[1]**, Kevin Fu**[1]**, Harish Ravichandar**[1]
[1]Georgia Institute of Technology, [2]University of Texas at Austin
Email: pierce.howell@gatech.edu

**Abstract:** Recent advances in multi-agent reinforcement learning (MARL) are enabling impressive coordination in heterogeneous multi-robot teams. However, existing approaches often overlook the challenge of generalizing learned policies to teams of new compositions, sizes, and robots. While such generalization might not be important in teams of virtual agents that can retrain policies on-demand, it is pivotal in multi-robot systems that are deployed in the real-world and must readily adapt to inevitable changes. As such, multi-robot policies must remain robust to team changes – an ability we call *adaptive teaming*. In this work, we investigate if *awareness and communication of robot capabilities* can provide such generalization by conducting detailed experiments involving an established multi-robot test bed. We demonstrate that shared decentralized policies, that enable robots to be both aware of and communicate their capabilities, can achieve adaptive teaming by implicitly capturing the fundamental relationship between collective capabilities and effective coordination. Videos of trained policies can be viewed at https://sites.google.com/view/cap-comm.

**Keywords:** Heterogeneity, Multi-Robot Teaming, Generalization

## 1 Introduction

Heterogeneous robot teams have the potential to address complex real-world challenges that arise in a wide range of domains, such as precision agriculture, defense, warehouse automation, supply chain optimization, and environmental monitoring. However, a key hurdle in realizing such potential is the challenge of ensuring effective communication, coordination, and control.

Existing approaches to address the challenges of multi-robot systems can be crudely categorized into two groups. First, classical approaches use well-understood controllers with simple local interaction rules, giving rise to complex global emergent behavior [1]. Indeed, such controllers have proven extraordinarily useful in diverse domains. However, designing them requires both significant technical expertise and considerable domain knowledge. Second, recent learning-based approaches alleviate the need for expertise and domain knowledge by leveraging advances in learning frameworks and computational resources. Learning has been successful in many domains, such as video games [2], autonomous driving [3], disaster response [4], and manufacturing [5].

However, learning approaches are not without their fair share of limitations. First, the majority of existing methods focus on homogeneous teams and, as such, cannot handle heterogeneous multi-robot teams. Second, and more importantly, even existing methods designed for heterogeneous teams are often solely concerned with the challenge of learning to coordinate a given team, entirely ignoring the challenge of *generalizing* the learned behavior to new teams. Given the potentially prohibitive cost of retraining coordination policies after deployment in real-world settings, it is imperative that multi-robot policies generalize learned behaviors to inevitable changes to the team.

In this work, we focus on the challenge of generalizing multi-robot policies to team changes. In particular, we focus on generalization of trained policies to teams of new compositions, sizes, and robots that are not encountered in training (see Fig. 1). We refer to such generalization as *adaptive teaming*, wherein the learning policy can readily handle changes to the team without additional

---

[*]Equal Contribution. [†] This work was supported in part by the Army Research Lab under Grants W911NF-17-2-0181 and W911NF-20-2-0036

7th Conference on Robot Learning (CoRL 2023), Atlanta, USA.

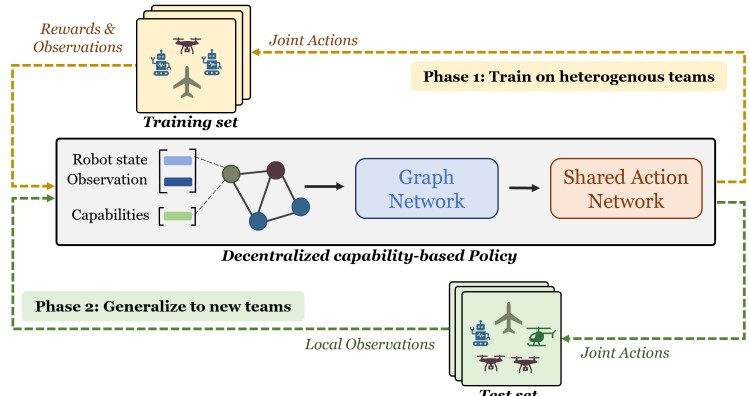

Figure 1: We investigate the role of capability awareness and communication in generalizing decentralized heterogeneous multi-robot coordination policies to teams of new composition, size, and robots.

training. To this end, we need policies that can reason about how a group of diverse robots can collectively achieve a common goal, without assigning rigid specialized roles to individual robots.

We investigate the role of *robot capabilities* in generalization to new teams. Our key insight is that adaptive teaming requires the understanding of how a team's diverse capabilities combine to dictate the behavior of individual robots. For instance, consider an autonomous heterogeneous team responding to multiple concurrent wildfires. Effective coordination in such situations requires reasoning about the opportunities and constraints introduced by the robots' individual and relative capabilities, such as speed, water capacity, and battery range. In general, robots must learn how their individual capabilities relate to those of others to determine their role in achieving shared objectives.

We develop a policy architecture that can explicitly reason about robot capabilities when selecting actions. Our architecture has four key properties: i) *capability awareness*: our design enables actions to be conditioned on continuous capabilities in addition to observations, ii) *capability communication*: we leverage graph networks to learn how robots must communicate their capabilities iii) *robot-agnostic*: we utilize parameter sharing and learn policies that are not tied to individual robots, and iv) *decentralized*: our trained policies can be deployed in a decentralized manner. Together, these four properties provide the potential to generalize to new teams. One can view this design as an extension of agent identification techniques [6] to the metric space of capabilities. As such, capabilities do not merely serve to distinguish between agents during training to enable behavioral heterogeneity [7], but also to provide a more general means to encode how individual and relative capabilities influence collective behavior.

We evaluate the utility of capability awareness and communication in two heterogeneous multi-robot tasks in sim and real. Our results reveal that both awareness and communication of capabilities can enable adaptive teaming, outperforming policies that lack either one or both of these features in terms of average returns and task-specific metrics. Further, capability-based policies achieve superior zero-shot generalization than existing agent identification-based techniques, while ensuring comparable performance on the training set.

## 2 Related Work

**Learning for multi-robot teams**: Recent advances in deep learning are providing promising approaches that circumvent the challenges associated with classical control of multi-robot systems. Multi-agent reinforcement learning (MARL), in particular, has been shown to be capable of solving a wide variety of tasks, including simple tasks in the multi-agent particle environments (MPE) [8], complex tasks under partial observability [9], coordinating an arbitrary number of agents in video games [2], and effective predictive modeling of multi-agent systems [10]. These approaches are driven by popular MARL algorithms like QMIX [11], MADDPG [8], and MAPPO [12] – nontrivial extensions of their single agent counterparts DQN [13], DDPG [14], and PPO [15], respectively. We adopt a PPO-based learning framework given its proven benefits despite its simplicity [12]. Centralized training, decentralized execution (CTDE) is a commonly used framework in which decentralized agents learn to take actions based on local observations while a centralized critic provides

feedback based on global information [16, 17]. We use the CTDE paradigm as it lends itself naturally to multi-robot teams since observation and communication are often restricted. However, its important to note that our approach is agnostic to the specific learning algorithm.

**Learning for heterogeneous teams**: Many MARL algorithms were originally designed for use in homogeneous multi-agent teams. However, truly homogeneous multi-robot teams are rare except because of manufacturing differences, wear and tear, or task requirements. Most real-world multi-robot problems such as search & rescue, agriculture, and surveillance require a diverse set of capabilities aggregated from heterogeneous robots [18–20]. While many MARL approaches consider heterogeneity, they either tend to focus on differences in behavior exhibited by physically identical robots [21], or identical behavior exhibited by physically-different robots [22, 23]. A common strategy used to elicit heterogeneous behavior from shared models is referred to as agent identification or behavioral typing, in which the agents' observations are appended with an agent-specific index [24, 25]. While these methods have been shown to be highly effective, recent investigations have revealed issues with scalability [26], and robustness to observation noise [7]. While capability-awareness is similar in spirit to existing identification-based techniques, it does not require assigning indices to individual robots and can thus generalize to teams with new robots. Further, most existing methods do not simultaneously handle teams with physical and behavioral differences. According to a recent heterogeneous multi-robot taxonomy [7], our work falls under the category consisting of physically-different robots that differ in behavior, but share the same objective. Two recent approaches belong to this same category [4, 7]. However, one is limited to a discrete set of robot types [4] and the other learned decentralized robot-specific policies that cannot handle the addition of new robots and might not generalize to new compositions [7].

**Generalization**: For applications to real-world multi-robot systems, it is essential to consider the generalization capabilities of learned control policies. In our formulation, there are two axes of generalization in heterogeneous multi-robot teams: combinatorial generalization (new team sizes and new compositions of the same robots) and individual capability generalization (new robots). Prior works reliant on feed forward or recurrent networks tend to be limited to teams of static size [8, 27]. Combinatorial generalization for homogeneous teams can be achieved with graph network-based policies [27, 28]. However, existing methods tend to struggle with generalization in the presence of heterogeneity [29]. While methods that employ agent identification [24, 25] might be able to achieve combinatorial generalization by reusing the IDs from training, it is unclear how they can handle new robots. In stark contrast, capability-based policies are robot-agnostic and can take the capabilities of the new robot as an input feature to determine its actions.

## 3 Capability Awareness and Communication for Adaptive Teaming

In this section, we first model heterogeneous teams and then introduce policy architectures that enable capability awareness and communication, along with the associated training pipeline.

### 3.1 Modeling Heterogeneous Multi-Robot Teams

We model teams of $N$ heterogeneous robots as a graph $\mathcal{G} = (\mathcal{V}, \mathcal{E})$, where each node $v_i \in \mathcal{V}$ is a robot, and each edge $e_{ij} = (v_i, v_j) \in \mathcal{E}$ is a communication link. We use $z_i$ to denote the observations of the $i$th robot, which includes its capabilities and its sensor readings of the environment. We assume that the robots' heterogeneity can be captured by their capabilities. We represent the capabilities of the $i$th robot by a real-valued vector $c_i \in \mathcal{C} \subseteq \mathbb{R}_+^C$, where $\mathcal{C}$ is the $C$-dimensional space of all capabilities of the robots. An example of a multi-dimensional capability is a vector with elements representing payload, speed, and sensing radius. When robot $i$ does not possess the $k$th capability, the $k$th element of $c_i$ is set to zero.

### 3.2 Problem Description

We are interested in learning a decentralized control policy that can i) effectively coordinate a team of heterogeneous robots to achieve the task objectives, and ii) generalize readily to teams of both novel compositions and novel robots that are not encountered during training. Our problem can be viewed as a multi-agent reinforcement learning problem that can be formalized as a decentralized partially-observable Markov Decision Process (Dec-POMDP) [30]. We expand on the Dec-POMDP formulation to incorporate the *capabilities* of heterogeneous robots and arrive at the tuple

$\langle \mathcal{D}, \mathcal{S}, \{\mathcal{A}_i\}, \{\mathcal{Z}_i\}, \mathcal{C}, T, R, O\rangle$ where $\mathcal{D}$ is the set of $N$ robots, $\mathcal{S}$ is the set of global states, $\{\mathcal{A}_i\}$ is a set of action spaces across all robots, $\{\mathcal{Z}_i\}$ is a set of joint observations across all robots, $\mathcal{C}$ is the multi-dimensional space of capabilities, $R$ is the global reward function, and $T$ and $O$ are the joint state transition and observation models, respectively. Our objective is to learn decentralized action policies that control each robot to maximize the expected return $\mathbb{E}[\sum_{t=0}^{T_h} r_t]$ over the task horizon $T_h$. The decentralized policy of the $i$th robot $\pi(a_i | \widetilde{o}_i)$ defines the probability that Robot $i$ takes Action $a_i$ given its effective observation $\widetilde{o}_i$. The effective observation $\widetilde{o}_i$ of the $i$th robot is a function of both its individual observation and that of others in its neighborhood.

### 3.3 Policy Architecture

To enable capability awareness and communication in multi-robot coordination policies, we designed a policy architecture that leverages graph convolutional networks (GCNs) since a plethora of recent approaches attest to their ability to learn effective communication protocols and enable decentralized decision-making in multi-agent teams [27]. Further, operations are local to nodes and can therefore generalize to graphs of any topology (i.e., permutation invariance [31]). We illustrate our architecture in Fig. 1 and explain its components below. Specific details including the exact architecture and hyperparameters we used can be found in Appendix E.

**Capability awareness**: We argue that the heterogeneity of robots can have a significant impact on how the team must coordinate to achieve a task. Specifically, individual and collective capabilities can affect the roles the robots play within the team. For instance, consider a heterogeneous mobile robot team responding to wildfire incidents at multiple locations. To effectively respond, a robot within the team must account for its speed and water capacity. Therefore, it is necessary that robots are aware of their capabilities. To enable such awareness, we append each robot's capability vector $c_i$ to its observations before passing them along as node features to the graph network. This information will help each robot condition its actions not just on observations, but also on its capabilities.

**Capability communication**: In addition to awareness, communicating capability information can enable a team to reason about how its collective capabilities impact task performance. Revisiting our wildfire example, robots in the response team can effectively coordinate their efforts by implicitly and dynamically taking on roles based on their relative speed and water capacity. But such complex decision making is only possible if robots communicate with each other about their capabilities. Each node in our GCN-based policy receives capability information along with the corresponding robot's local observations so the learned communication protocol can help the team communicate and effectively build representations of their collective capabilities.

Note that our policy is robot-agnostic and learns the implicit and interconnected relationships between the observations, capabilities, and actions of all robots in the team. Further, as we demonstrate in our experiments, capability awareness and communication enables generalization to teams with new robot compositions, sizes, and even to entirely new robots as long as their capabilities belong to the same space of capabilities $\mathcal{C}$.

### 3.4 Training Procedure

We utilize parameter sharing to train a single action policy that is shared by all of the robots. Parameter sharing is known to improve learning efficiency by limiting the number of parameters that must be learned. More importantly, parameter sharing is required for our problem so policies can transfer to new robots without the need for training new policies or assigning robots to already trained policies. Additionally, we believe that parameter sharing serves a secondary role in learning generalizable strategies for efficient generalization. Sharing parameters enables the policy to learn generalized coordination strategies that, when conditioned on robot capabilities and local observations, can be adapted to specific robots and contexts.

We employ a centralized training, decentralized execution (CTDE) paradigm to train the action policy. We apply an actor-critic model, and train using proximal policy optimization (PPO). The actor-critic model is composed of a decentralized actor network (i.e., shared action policy) that maps robots observations to control actions, and a centralized critic network [12], which estimates the value of the team's current state based on centralized information about the environment and robots aggregated from individual observations. Finally, we trained our policies on multiple teams until they converged, with the teams changing every 10 episodes to stabilize training.

# 4 Experimental Design

We conducted detailed experiments to evaluate how capability awareness and communication impact generalization to: i) new team sizes and compositions, and ii) new robots with unseen capabilities.

**Environments**: We designed two heterogeneous multi-robot tasks for experimentation:

- `Heterogeneous Material Transport (HMT)`: A team of robots with different material carrying capacities for lumber and concrete (denoted by $c_i \in \mathbb{R}^2$ for the $i$th robot) must transport materials from lumber and concrete depots to a construction site to fulfill a pre-specified quota while minimizing over-provision. We implemented this environment as a Multi-Particle Environment (MPE) [32] and leverage the infrastructure of EPyMARL [33].
- `Heterogeneous Sensor Network (HSN)`: A robot team must form a single fully-connected sensor network while maximizing the collective coverage area. The $i$th robot's capability $c_i \in \mathbb{R}$ corresponds to its sensing radius. We implemented this environment using the MARBLER [34] framework which enables hardware experimentation in the Robotarium [35], a well-established multi-robot test bed.

**Policy architectures**: In order to systematically examine the impact of capability awareness and communication, we consider the following policy architectures:

- `ID(MLP)`: Robot ID-based MLP
- `ID(GNN)`: Robot ID-based GNN
- `CA(MLP)`: Capability-aware MLP
- `CA(GNN)`: Capability-aware GNN without communication of capabilities,
- `CA+CC(GNN)`: Both capability awareness and communication.

The ID-based baselines stand in for SOTA approaches that employ behavioral typing to handle heterogeneous teams [24, 25], and, as such, question the need for capabilities. The MLP based baselines help us investigate the need for communication. Finally, the `CA(GNN)` enables communication of observations but does not does not communicate capability information.

**Metrics**: For both environments, we compare the above policies using *Average Return*: the average joint reward received over the task horizon (higher is better). Additionally, we use environment-specific metrics. In `HMT`, we terminate the episodes when the quotas for both materials are met. Therefore, we consider `Average Steps` taken to meet the quota (lower is better). For `HSN`, we consider *Pairwise Overlap*: sum of pairwise overlapping area of robots' coverage areas (lower is better).

**Training**: For each environment, we used five teams with four robots each during training. We selected the training teams to ensure diverse compositions and degree of heterogeneity. For `HSN`, we sampled robots' sensing radius from the uniform distribution $U(0.2, 0.6)$. For `HMT`, we sampled robots' lumber and concrete carrying capacities from the uniform distribution $U(0, 1.0)$. We also assigned each robot a one-hot ID to train ID-based policies. We trained each policy with 3 random seeds. We resampled robot teams every 10 episodes to stabilize training.

# 5 Results

Below, we report i) performance on the training team, ii) zero-shot generalization to new teams, and iii) zero-shot generalization to new robots with unseen values of capabilities for each environment.

## 5.1 Heterogeneous Material Transport

We first focus on the Heterogeneous Material Transport (`HMT`) environment.

**Performance on training set**: To ensure considering capabilities does not negatively impact training, we first evaluate trained policies on the training set in terms of average return and average steps (see Fig. 2). We find that all policies resulted in comparable average returns (Fig. 2 (a)). However, the average number of steps per episode better captures performance in `HMT`, since episodes terminate early when the quota is filled. Compared to agent-ID policies, capability-based policies took fewer steps per episode (Fig. 2 (b)) to achieve comparable rewards, suggesting that reasoning about capabilities can improve task efficiency performance.

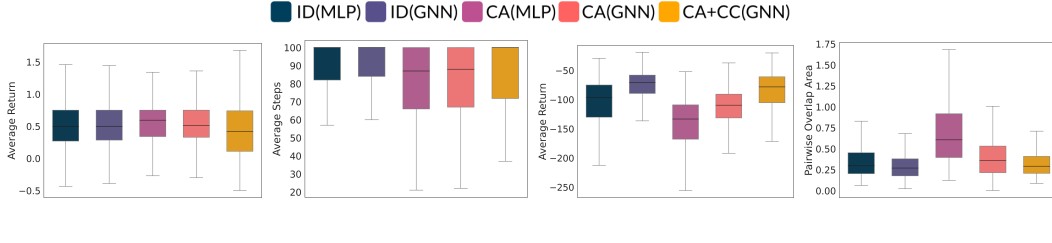

(a) HMT: Average return     (b) HMT: Average steps     (c) HSN: Average return     (d) HSN: Overlap area

Figure 2: When evaluated on *teams seen during training*, capability-aware policies performed comparably to ID-based policies in terms of both average return (higher is better) and task-specific metrics (lower is better).

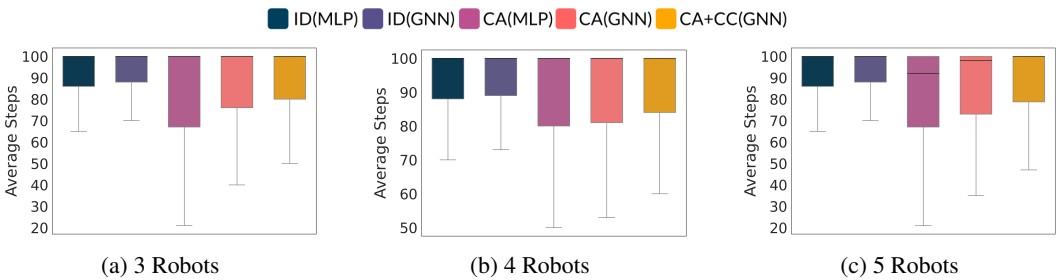

(a) 3 Robots        (b) 4 Robots        (c) 5 Robots

Figure 3: When generalizing to *new team compositions and sizes* in HMT, capability-based policies consistently outperformed ID-based policies in terms of average steps taken to meet the quota (lower is better).

**Zero-shot generalization to new team compositions and sizes**: We next evaluated how trained policies generalize to team compositions and sizes not encountered during training. Fig. 3 shows plots quantifying performance on new compositions with team sizes of 3, 4, and 5 robots. To ensure a fair comparison, we evaluated all policies on the same set of 100 teams by randomly sampling novel combinations of robots from the training set. We evaluated each policy on each test team across 10 episodes per seed. Given that this evaluation involved no new individual robots, we reused each robot's ID from the training set to facilitate the evaluation of ID-based policies.

We find that all capability-aware methods outperformed ID-based methods in return and task-specific metrics. This is likely due to capability-aware methods' ability to capture the relationship between robots' carrying capacities and the material quota. In contrast, ID-based methods must learn to implicitly reason about how much material each robot can carry. Interestingly, CA(MLP) resulted in fewer steps per episode (lowest mean and variance) across all team sizes, and outperformed both the other capability-based and communication-enabled baselines: CA(GNN) and CA+CC(GNN). This suggests that mere awareness of capabilities is sufficient to perform well in the HMT environment. Indeed, communication is not as essential in this task as robots can directly observe relevant information (e.g. material demands) and implicitly coordinate as long as they are aware of their own capabilities. It might be possible to further improve performance by learning to better communicate, but that would be significantly more challenging since the task can be mostly solved without communication.

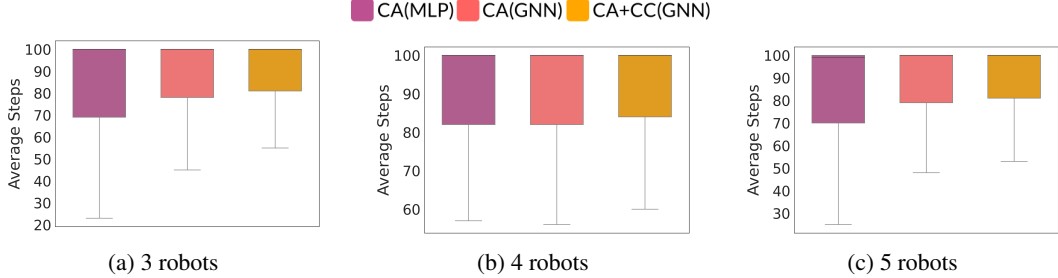

(a) 3 robots        (b) 4 robots        (c) 5 robots

Figure 4: When generalizing to *new robots with unseen values for capabilities* in HMT, policies that are only aware of capabilities (CA(MLP) and CA(GNN)) outperformed policies that also communicated capabilities (CA+CC(GNN)) in terms of average number of steps taken to transport the required material (lower is better).

**Zero-shot generalization to new robots**: In Fig. 4, we show the policies' ability to generalize to teams composed of new robots. Since the robots' capabilities in this evaluation are different from those of the robots in the training set, we could not evaluate agent-ID methods since there is no trivial way to assign IDs to the new robots. These results clearly demonstrate that reasoning about capabilities can enable generalization to teams with entirely new robots. Further, we again see that capability awareness without communication is sufficient to generalize in the HMT environment.

**Additional results**: We provide additional results for HMT by reporting more task-specific metrics and evaluations on significantly larger teams in Appendix A. The results on task-specific metrics further support the claim that capability-aware methods generalize better than ID-based methods. We also find that these benefits extend to teams consisting of 8, 10, and 15 robots. Taken together, the above results suggest that reasoning about capabilities (rather than assigned IDs) improves adaptive teaming, likely due to the ability to map capabilities to implicit roles.

## 5.2 Heterogeneous Sensor Network

Below, we discuss results on the Heterogeneous Sensor Network (HSN) environment.

**Performance on training set**: In Fig. 2 (c) and (d), we report the performance of trained policies in HSN on teams in the training set in terms of average return and pairwise overlap. All policies except CA(MLP) performed comparably and were able to effectively learn to maximize expected returns, achieve a fully connected sensor network, and minimize the pairwise overlap in coverage area. Further, CA+CC(GNN) and ID(GNN) perform similarly but marginally better than the other baselines. CA(MLP)'s suboptimal performance indicates that capability awareness in isolation without any communication hurts performance in the HSN task. Indeed, while it is possible to achieve good performance in HMT without communicating, HSN requires robots to effectively communicate and reason about their neighbors' sensing radii in order to form effective networks.

Taken together, these results suggest that capability awareness and communication can lead to effective training in heterogeneous teams. ID-based methods are able to perform at a similar level. This is to be expected given that we conducted these evaluations on the training set and IDs are sufficient to implicitly assign roles and coordinate heterogeneous robots within known teams.

**Zero-shot generalization to new team compositions and sizes**: In Fig. 5, we report the performance of the training policies in HSN when evaluated on teams of different compositions and sizes. We found that CA+CC(GNN) achieved the best average returns (highest mean and lowest variance) across all team sizes. Both ID-based methods (ID(MLP) and ID(GNN)) resulted in lower returns compared to all three capability-awareness baselines. Note that this is in stark contrast to the results for the training set in Fig. 2, demonstrating that IDs alone might help train heterogeneous teams but tend to generalize poorly to new team compositions and sizes. This is likely because ID-based policies fail to reason about robot heterogeneity, and instead overfit the relationships between robot IDs and behavior in the training set. Further, CA+CC(GNN) in particular consistently outperformed all other policies across metrics and variations, suggesting that both capability awareness and communication are necessary to enable generalization in HSN.

**Zero-shot generalization to new robots**: We evaluated the trained policies' ability to generalize to teams of different sizes which are composed of entirely new robots whose sensing radii are different from those encountered in training. Similar to HMT, we cannot evaluate ID-based policies on teams

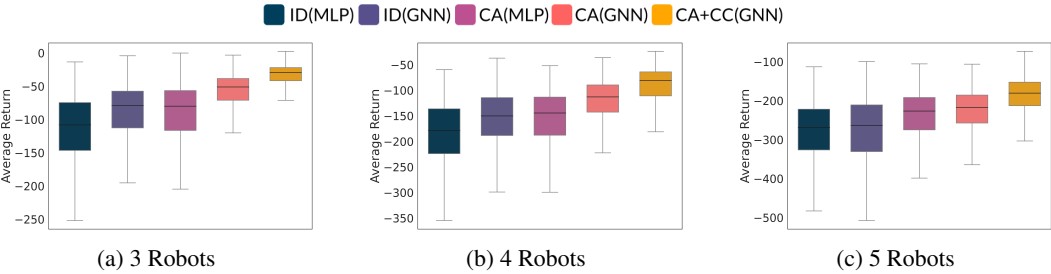

(a) 3 Robots           (b) 4 Robots           (c) 5 Robots

Figure 5: When generalizing to *new team compositions and sizes* in HSN, capability-based policies consistently outperformed ID-based baselines in terms of average return (higher is better). Further, combining awareness and communication of capabilities resulted in the best generalization performance.

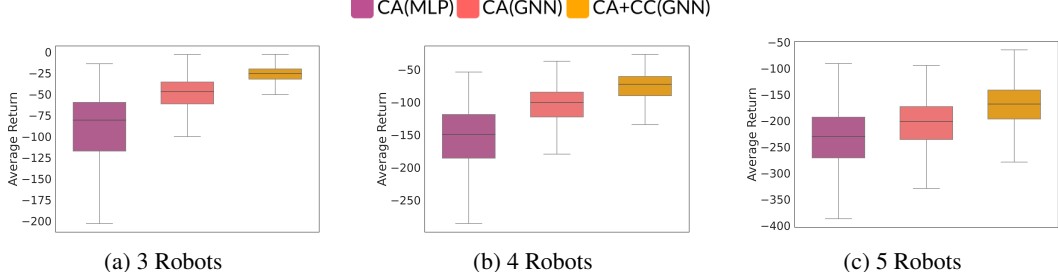

| (a) 3 Robots | (b) 4 Robots | (c) 5 Robots |

Figure 6: When generalizing to *teams comprised of new robots* in HSN, combining awareness and communication of capabilities (`CA+CC(GNN)`) achieves higher average returns than baselines that are merely aware of capabilities, irrespective of whether they communicate observations (`CA(GNN)`) or not (`CA(MLP)`).

with new robots since there is no obvious way to assign IDs to the new robots. In Fig. 6, we report the performance of all three capability-based policies in terms of average return. Both GNN-based policies (`CA(GNN)` and `CA+CC(GNN)`) considerably outperform the `CA(MLP)` policy, underscoring the importance of communication in generalization to teams with new robots. However, we also see that communication of observations alone is insufficient, as evidenced by the fact that `CA+CC(GNN)` (which communicates both observations and capabilities) consistently outperforms `CA(GNN)` (which only communicates observations).

**Real-robot demonstrations**: We also deployed the trained policies of `CA+CC(GNN)`, `CA(MLP)`, and `CA(GNN)` on the physical Robotarium (see Section B.1 for further details and snapshots). Overall, we find that the benefits reported above extend to physical robot teams. We find that `CA+CC(GNN)` and `CA(GNN)` policies generalize to physical robots and successfully build a sensor network while minimizing sensing overlap for teams of 3 and 4 robots. The `CA(MLP)` policy resulted in significantly worse performance, where robots' executed paths provoked significant engagement of the Robotarium's barrier certificates due to potential collisions.

**Additional results**: We provide additional results for HSN by reporting more task-specific metrics in Appendix B. Much like the results for HMT, the results on task-specific metrics further support the claim that capability-aware methods show superior adaptive teaming ability compared with ID-based methods. The additional results also support our claim in this section that communication of capabilities is essential for success on this task.

## 6 Limitations

While our framework could reason about many different capabilities simultaneously, our experiments only involved variations in 1-D and 2-D capabilities. We also only consider generalization to new *values* for capabilities; we do not consider generalization to new types of capabilities. Additionally, our work only considers the representation of robot's capabilities that we can quantify. Handling implicit capabilities and communication thereof may benefit from additional meta-learning mechanisms, uncovering a more general relationship between robots' learned behaviors and capabilities. Further, we do not consider high-level planning and task-allocation and rely solely on the learning framework to perform implicit assignments to sub-tasks within the macro task. Future work can investigate appropriate abstractions and interfaces for considering both learning-based low-level policies and efficient algorithms for higher-level coordination. Lastly, we only considered fully-connected communication graphs in our evaluations for simplicity. While graph networks are known to effectively share local observations for global state estimations in partially-connected teams [27, 36], it is unclear if such ability will translate to the communication of capabilities.

## 7 Conclusion

We investigated the utility of awareness and communication of robot capabilities in the generalization of heterogeneous multi-robot policies to new teams. We developed a graph network-based decentralized policy architecture based on parameter sharing that enables robots to reason about and communicate their observations and capabilities to achieve adaptive teaming. Our detailed experiments involving two heterogeneous multi-robot tasks unambiguously illustrate the importance and the need for reasoning about capabilities as opposed to agent IDs.

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

# A    Heterogeneous Material Transport (HMT) Additional Results

This section provides additional results for the HMT environment. Specifically, we provide additional task-specific metrics for the generalization experiments, and new generalization results for robot teams of significantly larger sizes (i.e. team sizes of 8, 10, and 15 robots).

The task-specific metrics defined below evaluate the rate at which each policy contributes to fulfilling the total quota, and the individual quotas for lumber and concrete:

- % of episodes by which the total quota was filled.
- % of lumber quota remaining.
- % of concrete quota remaining.

For both generalization to new teams (see Fig. 7) and new robots (see Fig. 8), capability-aware methods filled the total quota in fewer episode steps compared to the ID-based methods, while generally better preventing over-provisioning of both lumber and concrete. This result further supports our claim that capability awareness improves generalization performance. Observing Fig. 9, we find that these benefits of capability-based policies extend to considerably larger team sizes.

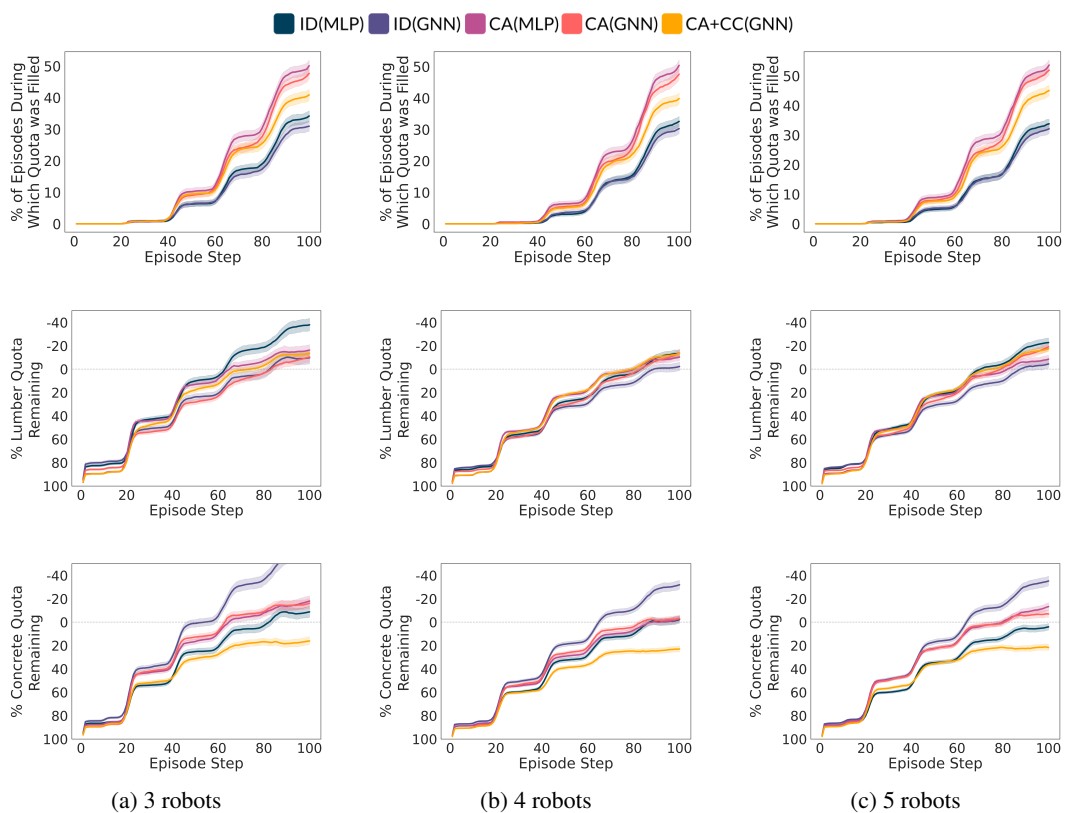

Figure 7: Policies with capability awareness outperform agent ID methods at meeting the material quota with a minimal number of steps when generalizing to new team compositions. Capability-awareness methods without communication of capabilities (i.e. CA(MLP) & CA(GNN)) outperform methods with capability communication for this task.

# B    Heterogeneous Sensor Network (HSN) Additional Results

This section provides additional results on i) training performance and ii) task-specific metrics.

The task-specific metrics for the HSN environment are the following:

- Pairwise overlap: The sum of pairwise overlap in coverage area among robots (lower is better).

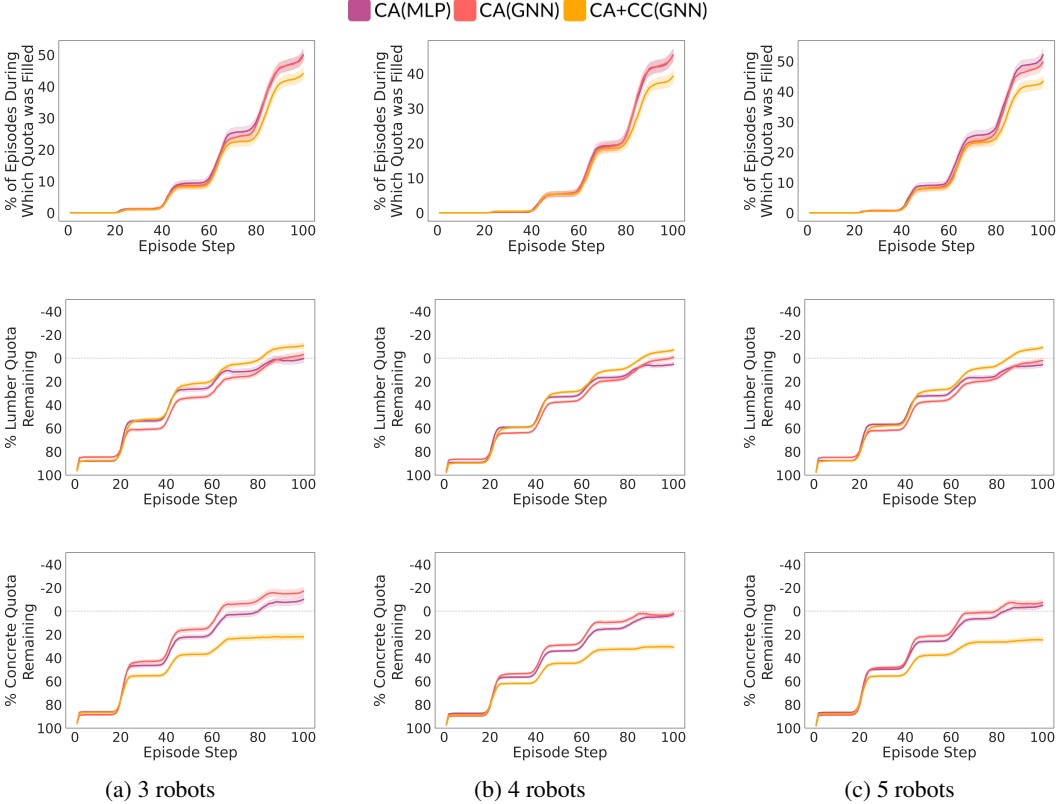

Figure 8: Policies without communication of capability-awareness (i.e. `CA(MLP)` and `CA(GNN)`) outperformed the policy with communication of capabilities (`CA+CC(GNN)` on task-specific metrics when generalizing to new robots with capabilities not seen during training.

- % of fully connected teams (by episode step): Percentage of teams that managed to form a sensor network that connects all of the robots (higher is better).

In Fig. 12, we report the training performance (i.e. training teams only) for each policy. The training curve suggest that capability-aware and ID-based methods perform comparably during learning. Notably, the communication models `ID(GNN)` and `CA+CC(GNN)` converge faster and achieve higher overall returns than other methods. This result suggests that communication between robots significantly assists in learning collaborative behavior.

Capability-aware policies again demonstrate superior performance when generalizing to new teams (see Fig. 10) and new robots (see Fig. 11) on task-specific metrics, highlighting the importance of capability awareness for generalization to robot teams with new robots, team sizes, and team compositions. Notably, `CA+CC(GNN)` results in significantly lower pairwise overlap for robot teams of size 3 and 4 robots, and marginally lower pairwise overlap for robot teams of size 5, compared to ID-based methods. This suggests the communication-enabled policy effectively learns to communicate capabilities and that such communication of capabilities is essential for generalization in this task.

## B.1 Images of Robotarium Experiments

In this section, we present visual representations derived from actual robot demonstrations of the trained capability-aware communication policy. Videos of the robot demonstrations can be found at: `https://sites.google.com/view/cap-comm`.

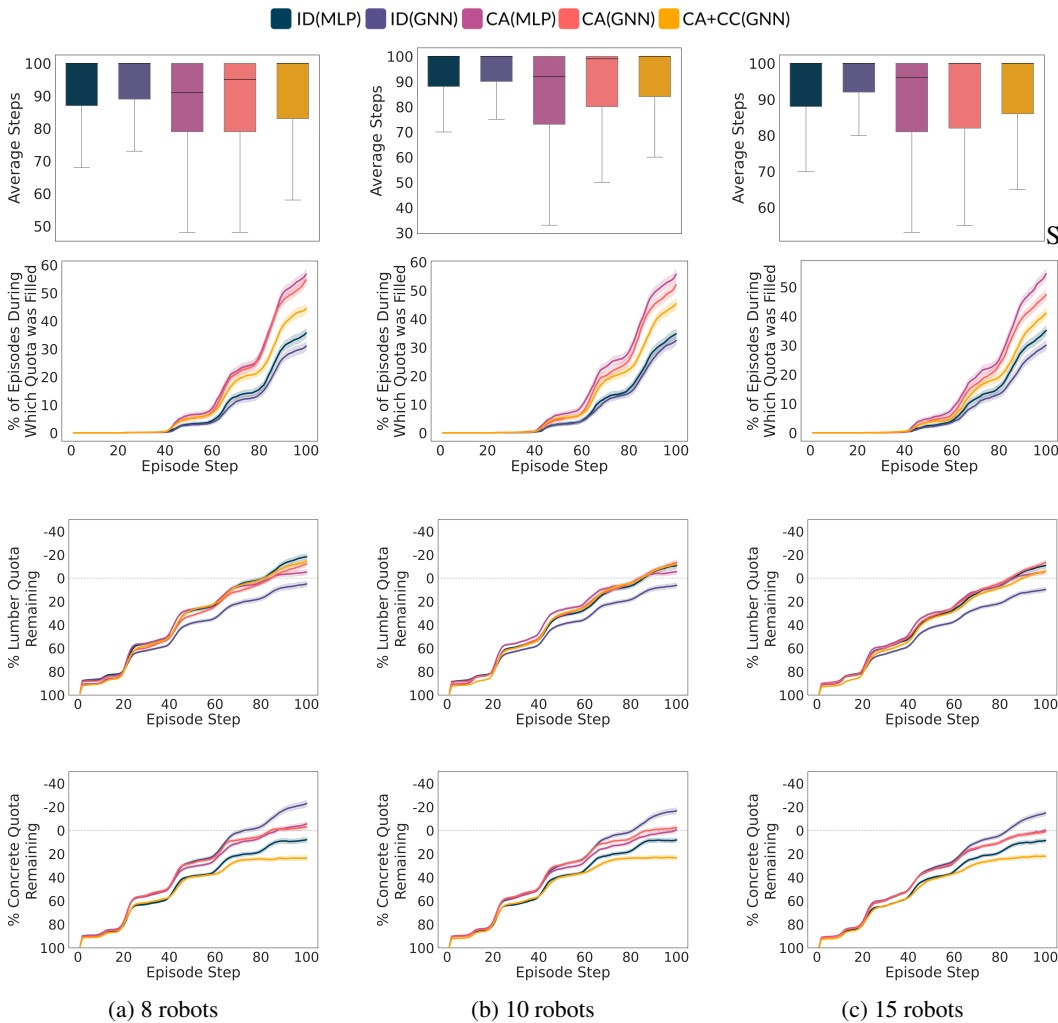

Figure 9: Experiments evaluating the generalization of policies to significantly larger teams (size 8, 10, and 15) compared to the training team size (size 4). Policies with capability awareness outperform agent ID methods at meeting the total material quota with a minimal number of steps when generalizing to new, large team compositions. Capability-awareness methods without communication of capabilities (i.e. CA(MLP) & CA(GNN)) outperform methods with capability communication for this task.

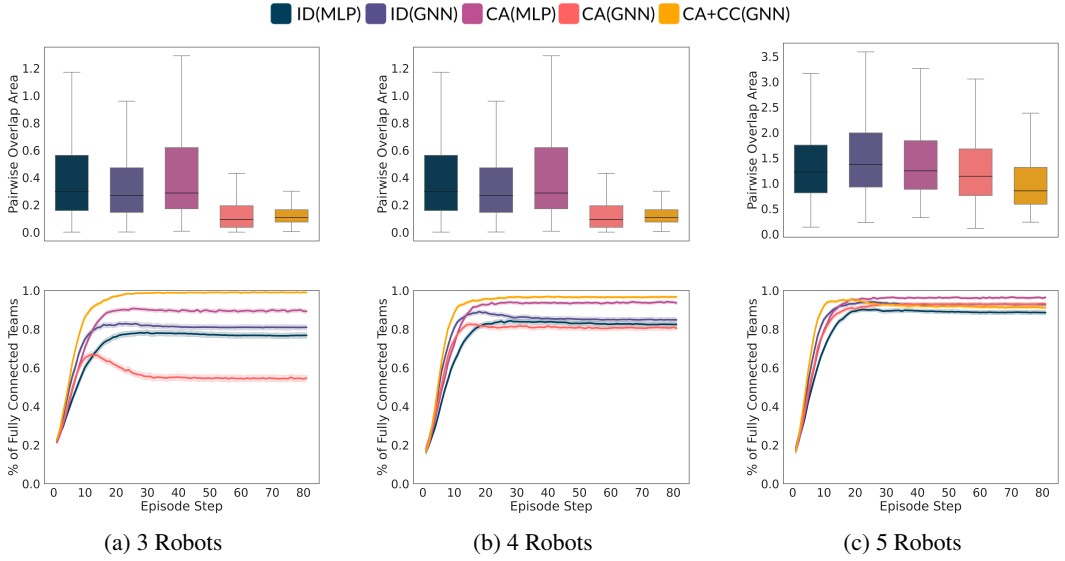

(a) 3 Robots

(b) 4 Robots

(c) 5 Robots

Figure 10: Capability-based policy architectures consistently outperform ID-based baselines both in terms of average return and task performance metrics when generalizing to new team compositions and sizes. Further, combining awareness and communication of capabilities results in the best generalization performance.

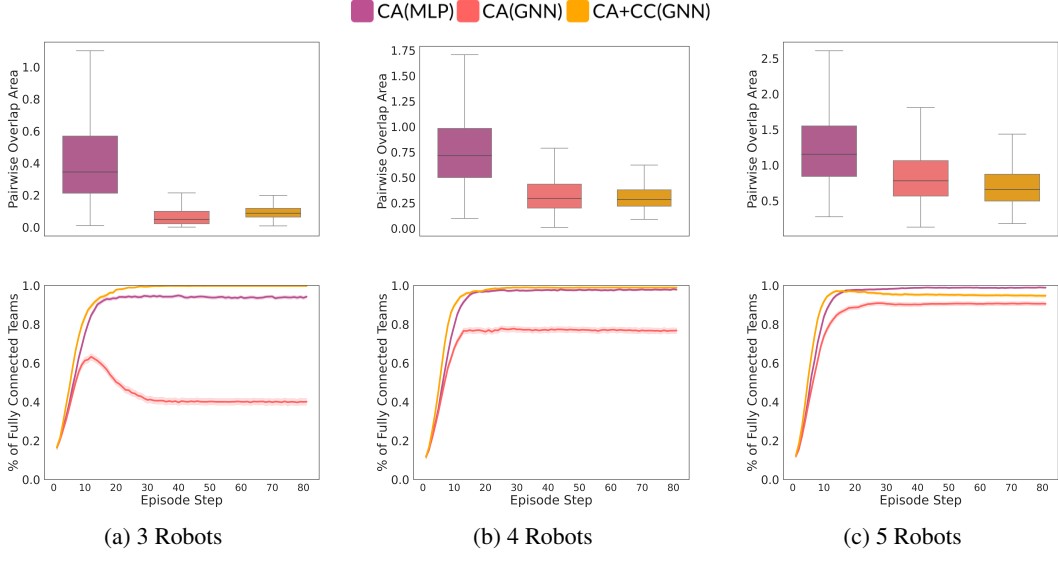

(a) 3 Robots

(b) 4 Robots

(c) 5 Robots

Figure 11: Policy architecture that combines awareness and communication (`CA+CC(GNN)`) of capabilities outperforms both other baselines (`CA(GNN)` in terms of % fully connected and `CA(MLP)` in terms of average return and pairwise overlap) when generalizing to teams comprised of new robots.

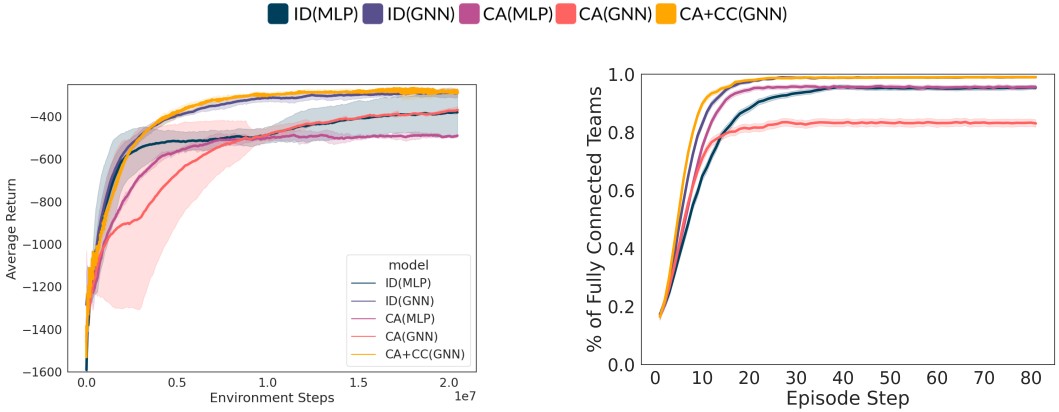

Figure 12: For the `HSN` environment, capability-aware policies perform comparably to ID-based policies in terms of training efficiency (first) and in terms of task-specific metrics when evaluating the trained policy on the training set.

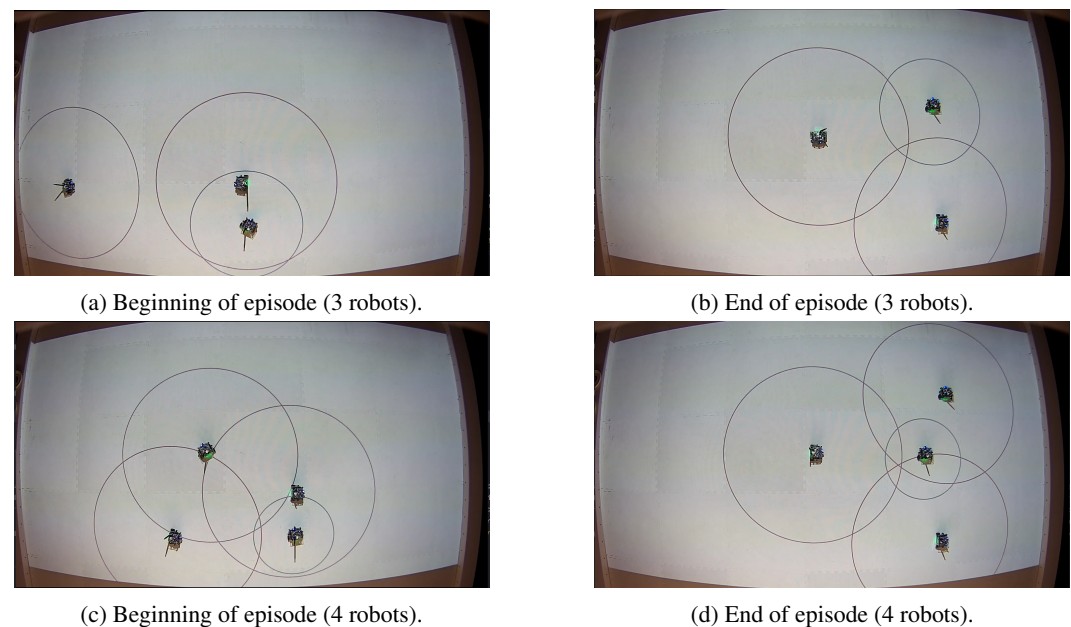

(a) Beginning of episode (3 robots).

(b) End of episode (3 robots).

(c) Beginning of episode (4 robots).

(d) End of episode (4 robots).

Figure 13: Demonstrations of `CA+CC(GNN)` policy deployed to real robot teams in the Robotarium testbed for the `HSN` task. See https://sites.google.com/view/cap-comm for videos of deployment to the Robotarium.

## C Environment Specifications

### C.1 Heterogeneous Material Transport

This section describes additional details about the heterogeneous material transport (HMT) environment.

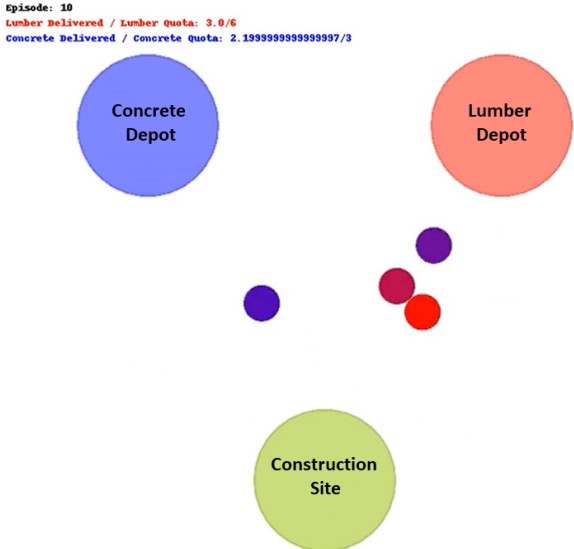

Figure 14: In the Heterogeneous Material Transport (HMT) environment, each agent's color is a mixture of blue and red, which represents its bias towards its carrying capacity for either lumber (red) or concrete (blue). The objective of the team is to fill the lumber and concrete quota at the construction site without delivering excess.

The lumber and concrete quota limit for the HMT environment are randomly initialized to an integer value between $(0.5 \times \text{n\_agents})$ and $(2.0 \times \text{n\_agents})$.

Robots have five available actions: they can move left, right, up, down, or stop. At the beginning of each episode, all of the robots begin at a random position in the *construction site* zone (see Fig. 14). The observation space for a robot is the combination of the robot's state and the environment's state: specifically it is composed of the robot's position, velocity, amount of lumber and concrete it's carrying, and its distance to each depot, the total lumber quota, the total concrete quota, the total amount of lumber delivered, and the total amount of concrete delivered. Robots' observations do not contain state information about other robots. Finally, we append the robot's unique ID for the ID baseline methods and the robot's maximum lumber and concrete carrying capacity for the CA methods to the robots' observations.

The total reward for each robot in HMT is computed by summing the individual rewards of each robot. Robots are rewarded when they make progress in meeting the lumber and concrete quotas and are penalized when they exceed the quota. If a robot enters the *lumber depot* or *concrete depot*, and the robot is empty (i.e. not loaded with any lumber or concrete), and the quota has yet to be filled, then the robot is rewarded with *pickup reward* of $0.25$. If the robot is loaded with material, then the robot is rewarded or penalized when it drops off the material at the *construction site*. Specifically, when a robot delivers a material and the quota for that material has yet to be filled, then the robot is rewarded with a positive *dropoff reward* of $0.75$. However, if the robot delivers a material and goes over the quota, then the robot is penalized with a negative *surplus penalty reward* proportional to the amount of surplus: $-0.10 \times$ surplus material. Finally, robots received a small *time penalty* of $-0.005$ for each episode step in which the total quota is not filled; this promotes the robots to finish the task as quickly as possible.

### C.2 Heterogeneous Sensor Network

This section describes additional details about the heterogeneous sensor network (HSN) environment.

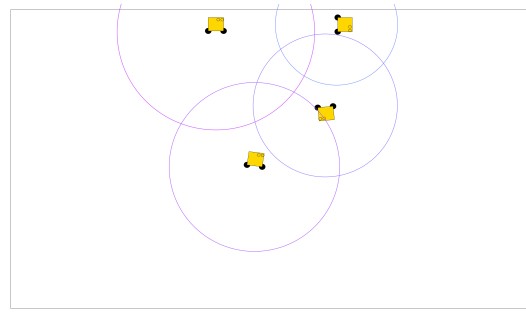
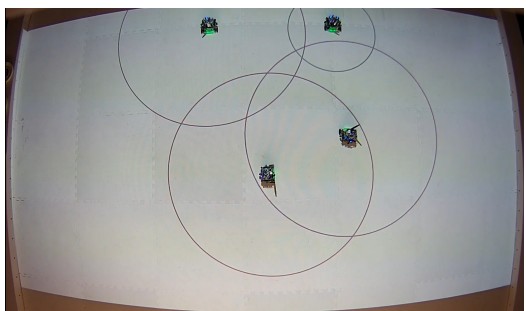

(a) Our agents running in simulation using the MAR-BLER framework.

(b) Our agents running in the physical Robotarium.

Robots have five available actions: they can move left, right, up, down, or stop. After selecting an action, the robots move in their selected direction for slightly less than a second before selecting a new action. The robots start at random locations least 30cm apart from each other, move at ~21cm/second, and utilize barrier certificates [35] that takes effect at 17cm away to ensure they do not collide when running in the physical Robotarium.

The reward from the heterogeneous sensor network environment is a shared reward. We describe the reward below:

$$D(i,j) = ||p(i) - p(j)|| - (c_i + c_j)$$

$$r(i,j) = \begin{cases} -0.9 * |D(i,j)| + 0.05, & \text{if } D(i,j) < 0 \\ -1.1 * |D(i,j)| - 0.05, & \text{otherwise} \end{cases}$$

$$R = \sum_{i<j}^{N} r(i,j)$$

where $i$ and $j$ are robots, $p(i)$ is the position of robot $i$, $c_i$ is the (capability) sensing radius of robot $i$, and $R$ is the cumulative team reward shared by all the robots. The above reward is designed to reward the team when robots connect their sensing regions while minimizing overlap so as to maximize the total sensing area.

## D  Training and Evaluation Specifications

This section describes the design of the training teams and the sampling of evaluation teams for both environments. To learn generalized coordination behavior, the training teams were required to be diverse in terms of composition and capture the underlying distribution of robot capabilities.

### D.1  Heterogeneous Material Transport

| Training Team Number | (concrete capacity, lumber capacity) |
|:---:|:---:|
| 1 | $(0.9, 0.1), (0.7, 0.3), (1.0, 0.0), (0.0, 1.0)$ |
| 2 | $(0.9, 0.1), (0.7, 0.3), (0.0, 1.0), (0.2, 0.8)$ |
| 3 | $(0.8, 0.2), (0.3, 0.7), (0.4, 0.6), (0.7, 0.3)$ |
| 4 | $(1.0, 0.0), (0.0, 1.0), (0.1, 0.9), (0.3, 0.7)$ |
| 5 | $(0.6, 0.4), (0.3, 0.7), (0.7, 0.3), (0.0, 1.0)$ |

Table 1: Training teams used for the HMT task (five teams of four robots each).

### D.2  Heterogeneous Sensor Network

To design these training teams, we first binned robot capabilities into small, medium, and large sensing radii with bin ranges $[0.2m, 0.33m]$, $[0.33m, 0.46m]$, and $[0.46m, 0.60m]$ respectively. We then generated all possible combinations with replacements for teams composed of four robots of small, medium, and large robots for a total of 15 teams. Each robot assigned to one of the bins small, medium, and large had its capability (i.e. sensing radius) uniformly sampled within the bin

range. This resulted in 15 total teams, for which we hand-selected 5 sufficiently diverse teams to be the training teams. The resulting training teams are given in Table 2.

| Training Team Number | Robot Sensing Radii ($m$) |
|---|---|
| 1 | $(0.2191), (0.2946), (0.2608), (0.3668)$ |
| 2 | $(0.2746), (0.2746), (0.5824), (0.5756)$ |
| 3 | $(0.3178), (0.3467), (0.5317), (0.6073)$ |
| 4 | $(0.2007), (0.5722), (0.5153), (0.4622)$ |
| 5 | $(0.4487), (0.5526), (0.5826), (0.58343)$ |

Table 2: Training teams used for the HSN task (five teams of four robots each).

The evaluation robot teams were sampled differently for the different experimental evaluations performed. In the training evaluation experiment, the teams were the same as the training teams in Table 2. Teams for the generalization experiment to new team compositions, but not new robots, were sampled randomly from the 20 robots from the training teams (with replacement). Each robot from the pool of 20 robots was sampled with equal probability. In contrast, teams for the generalization experiment to new robots were generated by randomly sampling new robots, where each robot's sensing radius was sampled from a uniform distribution independently $U(0.2m, 0.6m)$. For the two generalization experiments, 100 total teams were sampled. Each algorithm was evaluated on the same set of sampled teams by fixing the pseudo random number generator's seed.

We first focus on the training curves and subsequent evaluations conducted on the training set, without considering generalization. The goal of this experiment is to ensure introducing capabilities does not negatively impact training. The learning curves in terms of average return are shown in Fig. 12. All models achieved convergence within 20 million environment steps, with ID(GNN) and CA+CC(GNN) exhibited both the fastest convergence and the highest returns. These results suggest that communication of individual robot features, whether based on IDs or capabilities, improves learning efficiency and performance for heterogeneous coordination.

## E  Policy Details

### E.1  Graph Neural Networks

We employ a graph convolutional network (GCN) architecture for the decentralized policy $\pi_i$, which enables robots to communicate for coordination according to the robot communication graph $\mathcal{G}$.

A GCN is composed of $L$ layers of graph convolutions, followed by non-linearity. In this work, we consider a single graph convolution layer applied to node $i$ is given by

$$h_i^{(l)} = \sigma \left( \sum_{j \in \mathcal{N}(i) \cup i} \phi_\theta(h_j^{(l-1)}) \right)$$

where $h_j^{(l-1)} \in \mathbb{R}^F$ is the node feature of node $j$, $\mathcal{N}(i) = \{j | (v_i, v_j) \in \mathcal{E}\}$ are all nodes $j$ connected to $i$, $\phi_\theta$ is node feature transformation function with parameters $\theta$, $\sigma$ is a non-linearity (e.g. Relu), and $h_i^l \in \mathbb{R}^G$ is the output node feature.

### E.2  Policy Architectures

Each of the graph neural networks in the GNN-based policy architectures evaluated are composed of an input encoder network, a message passing network, and an action output network. The encoder network is a 2-layer MLP with hidden dimensions of size 64. For the message passing network, a single graph convolution layer composed of 2-layer MLPs with ReLU non-linear activations. The action output network is additionally a 2-layer MLP with hidden dimensions of size 64. The learning rate is 0.005.

MLP(ID)/MLP(CA): The MLP architectures compose of a 4-layer multi-layer perceptron with 64 hidden units at each layer and ReLU non-linearities.

CA(GNN)/CA+CC(GNN)/ID(GNN): Each of the graph neural networks compose of an input "encoder" network, a message passing network, and an action output network. The encoder network and the

action output network are multi-layer perceptrons with hidden layers of size 64, ReLU non-linear activations, and with one and two hidden layers respectively. The message passing network is a graph convolution layer wherein the linear transformation of node features (i.e. observations) is done by a 2-layer MLP with ReLU non-linear activations and 64 dimensional hidden units, followed by a summation of the transformed neighboring node features. The ouptut node features a concatenated with the output feature from the encoder network. This concatenated features is the input to the two out action network. The `CA(GNN)` network doesn't communicate the robot's capabilities with the graph convolution layers. Rather, the capabilities are appended to the output of the encoder network and output of node features of the graph convolution layer just before the the action network. Thus, the the action network is the only part of this model that is conditioned on robot capabilities.

### E.3 Policy Training Hyper parameters

We detail the hyperparameters used to train each of the policies using proximal policy optimization (PPO) [37] in Table 3.

| Hyperparameter | Value |
|---|---|
| Action Selection (Training) | `soft action selection` |
| Action Selection (Testing) | `hard action selection` |
| Critic Network Update Interval | 200 steps |
| Learning Rate | 0.0005 |
| Entropy Coefficient | 0.01 |
| Epochs | 4 |
| Clip | 0.2 |
| Q Function Steps | 5 |
| Buffer Length | 64 |
| Number of training steps | $40 \times 10^6$ (HMT), $20 \times 10^6$ (HSN) |

Table 3: Hyperparameters used to train each of the policies with PPO.

