# OpenReview forum: "Generalization of Heterogeneous Multi-Robot Policies via Awareness and Communication of Capabilities"
_robot-learning.org/CoRL/2023/Conference — CoRL 2023 Poster_

### Official Review · Reviewer_jEiU · 2023-07-11

**Confidence:** 4
**Originality:** Very Good
**Technical Quality:** Fair
**Clarity Of Presentation:** Very Good
**Impact:** 2

**Recommendation:**

Weak Accept: I recommend accepting the paper, but will not argue for my recommendation if the majority of other reviewers have a different opinion.

**Review:**

Strengths

-  The paper identifies a learning problem of interest to the CoRL community.
-  The proposed method is tested on hardware, using the Robotarium multi-robot test bed.
-  The proposed method is observed to outperform current methods that are capabilities *un*aware.

Weaknesses

-  I would elaborate in the introduction what is the role of the task at hand in the learning process, and its impact on the generalization capacity of the learned policies:  How does the generalization capacity get affected when the task changes, for example, from collaborative mapping to multi-target tracking?  How do the robots can "link" the current awareness of the robots' capabilities to the needed capability compositions for the new task?

Answering the above questions may be outside the scope of the paper.  If this is the case, then I would state early on in the paper (or in the limitations) that the focus of the paper is to achieve generalization given a task, and not across tasks.  I believe such a clarification will help readers to understand the paper's scope.

-  In my opinion, additional experiments, across a wider range of team sizes and tasks, could strengthen the paper's conclusions about the generalization capacity of the proposed algorithm.  In more detail:

(i) Fig. 3, which is used to conclude that awareness and communication of capabilities do not hurt performance, is based on experiments over a single task that involves 4 robots only.  In my opinion, additional experiments, across different tasks and more robots, may be needed to substantiate the conclusion that awareness and communication of capabilities do not hurt performance.  Or, could the paper be revised to elaborate on why the presented set of experiments is enough to support the conclusion?

(ii) Fig. 4 demonstrates a nice generalization to robot teams of 3, 4, and 5 robots, given training on teams of 4 robots.  In my opinion, a wider range of team sizes could be used to demonstrate the generalization capacity of the proposed method (e.g., up to 10 or more robots per team).

**Quality Of The Limitations Section:**

Additional details required

**Questions For Rebuttal:**

Please see all my comments in the weakness section above, repeated here:


-  I would elaborate in the introduction what is the role of the task at hand in the learning process, and its impact on the generalization capacity of the learned policies:  How does the generalization capacity get affected when the task changes, for example, from collaborative mapping to multi-target tracking?  How do the robots can "link" the current awareness of the robots' capabilities to the needed capability compositions for the new task?

Answering the above questions may be outside the scope of the paper.  If this is the case, then I would state early on in the paper (or in the limitations) that the focus of the paper is to achieve generalization given a task, and not across tasks.  I believe such a clarification will help readers to understand the paper's scope.

-  In my opinion, additional experiments, across a wider range of team sizes and tasks, could strengthen the paper's conclusions about the generalization capacity of the proposed algorithm.  In more detail:

(i) Fig. 3, which is used to conclude that awareness and communication of capabilities do not hurt performance, is based on experiments over a single task that involves 4 robots only.  In my opinion, additional experiments, across different tasks and more robots, may be needed to substantiate the conclusion that awareness and communication of capabilities do not hurt performance.  Or, could the paper be revised to elaborate on why the presented set of experiments is enough to support the conclusion?

(ii) Fig. 4 demonstrates a nice generalization to robot teams of 3, 4, and 5 robots, given training on teams of 4 robots.  In my opinion, a wider range of team sizes could be used to demonstrate the generalization capacity of the proposed method (e.g., up to 10 or more robots per team).

**Robotics Focus:**

Sufficient demonstration on hardware

**Summary Of Paper:**

The paper investigates whether communication of robot capabilities can improve the generalization of RL-based multi-robot policies to different robot-team sizes and robot-type compositions.  To this end, the paper develops a decentralized policy architecture with parameter sharing that allows robots to use their capabilities as implicit role indicators within the multi-robot team.  The role of the policy is to enable the robots to reason about individual robots' capabilities and team capabilities. The proposed approach is evaluated in both simulator and hardware experiments of active sensing tasks with multiple robots.

**Summary Of Recommendation:**

The paper proposes an excellent problem and a method that appears promising. In my opinion, the current version of the paper would benefit from additional experiments, across a wider range of team sizes and tasks, to better support the paper's conclusions about the generalization capacity of the proposed algorithm.

UPDATE: In light of the experimental results provided during the rebuttal process, which they show that both capability and communication awareness may not necessarily increase perfromance, I keep the "Weak Accept" recommendation.  I look forward to future iterations of the proposed algorithm.

---

### Official Review · Reviewer_FZZg · 2023-07-18

**Confidence:** 4
**Originality:** Fair
**Technical Quality:** Good
**Clarity Of Presentation:** Very Good
**Impact:** 3

**Recommendation:**

Weak Accept: I recommend accepting the paper, but will not argue for my recommendation if the majority of other reviewers have a different opinion.

**Review:**

Strengths:
* The paper is overall clearly written.
* Improving generalization across heterogeneous robots is an important topic.
* The experiments are pretty comprehensive, including real robot results from Robotarium.

Weaknesses:
* As mentioned in the limitation section, only one dimension of robot capability is tested.
* The figures in the experiments need improvement - see below for details.
  * The legends and labels in Figure 3,4,5 are way too small.
  * How is the variance/std calculated in the figures?
  * Figures should be placed closer to the text that describes the corresponding figures.


**Quality Of The Limitations Section:**

Limitations are addressed clearly

**Questions For Rebuttal:**

Please improve the figures and include additional experiments on more than one dimension of robot capabilities.

**Robotics Focus:**

Sufficient demonstration on hardware

**Summary Of Paper:**

The paper focuses on improving the generalization of multi-agent RL in terms of different team compositions, team sizes, and different robot capabilities. The author proposes to include robots’ capability in the observation space and use graph networks to enhance communication. The experiments show that the proposed method is effective in improving the generalization of the baselines.

**Summary Of Recommendation:**

The paper presents a method for improving generalization in multi-agent RL. The contribution is relatively minor but it does show improvement against reasonable baselines.

---

### Official Review · Reviewer_foRa · 2023-07-20

**Confidence:** 4
**Originality:** Good
**Technical Quality:** Very Good
**Clarity Of Presentation:** Excellent
**Impact:** 3

**Recommendation:**

Weak Accept: I recommend accepting the paper, but will not argue for my recommendation if the majority of other reviewers have a different opinion.

**Review:**

Strengths:
- The paper is very well-written and exceptionally clear.
- The motivation is compelling (generalizing to heterogeneity in multi-robot teams without re-training).
- The evaluation is thorough, with systematic investigation of the impact on training and generalization at test-time and a thorough set of baselines.
- Quantitative results are compelling, showing clear benefits of the proposed method.
- Qualitative results are compelling, showing real robots performing the task in accordance with intuition (maximizing surface area while maintaining connection).

Weaknesses:
- The main weakness is that the proposed approach has limited technical novelty. If I understand correctly, one could argue that the algorithm is equivalent to simply augmenting the robot state with the “capability” information. It is then not surprising that critical state information (the sensing radius in the experiment) is instrumental in solving the multi-agent task, and that the learned policy can generalize to different values of the augmented state. Treating this information as “capability” rather than state seems like mere semantics.
- The task is relatively simple in nature and in a single domain. As the authors acknowledge in the limitations, they only test the case where the capability vector is a scalar. The robots are also all identical in the experiment (in terms of hardware rather than capability).
- It seems that while the policy is able to generalize to new *values* of capabilities, it’s unable to generalize to entirely new capabilities, since the capability space C must be known beforehand.

Minor notes:
- “Heterogeneity” is later defined as variation in robot behavior and hardware, but it could be helpful to define this term upfront in the introduction.
- Some additional experiments that are not necessary but could be helpful include (1) ablations justifying the CTDE paradigm and (2) testing the impact of variations in the communication graph structure.

**Quality Of The Limitations Section:**

Limitations are addressed clearly

**Questions For Rebuttal:**

Please see weaknesses above, in particular the first bullet point.

**Robotics Focus:**

Sufficient demonstration on hardware

**Summary Of Paper:**

This paper presents a new algorithm for multi-agent reinforcement learning. The authors propose adding the “capability” of each robot as additional information in MARL, so that the system has capability awareness (each robot knows its own capability) and capability communication (each robot knows about the capabilities of the team). The authors show that this approach significantly improves team performance in a real-world multi-robot test bed (the Robotarium) when generalizing to different team compositions, robot capabilities, and numbers of robots.

**Summary Of Recommendation:**

**Post-Rebuttal Update**: I am keeping my score as a Weak Accept. The authors' rebuttal acknowledges the lack of algorithmic novelty w.r.t. augmented state information.

The paper is well-executed, well-written, and technically sound, but I would like the authors to clarify how their approach differs from augmented state information.

---

### Official Review · Reviewer_TgTL · 2023-07-20

**Confidence:** 3
**Originality:** Good
**Technical Quality:** Good
**Clarity Of Presentation:** Very Good
**Impact:** 3

**Recommendation:**

Weak Accept: I recommend accepting the paper, but will not argue for my recommendation if the majority of other reviewers have a different opinion.

**Review:**

Strengths:
- The paper is written clearly.
- The experiment baselines are thorough
- The proposed method is sound, interesting, and to my knowledge this is a novel application to multi-robot policies.

Weaknesses:
- The results are only on one kind of task, where only one of the capabilities matters. While this is a good proof of concept, it would be more impactful to push the method further with more complex task/capability relationships.


**Quality Of The Limitations Section:**

Limitations are addressed clearly

**Questions For Rebuttal:**

- The graph edges are "communication links". What does this mean in practice? How does performance scale with graph connectivity?


Nits:
- Figure 3 is difficult to read.

**Robotics Focus:**

Sufficient demonstration on hardware

**Summary Of Paper:**

The paper tackles the problem of training policies that control multiple robots simultaneously, and is specifically interested in the case of 0-shot generalization to novel robot team configurations. The proposed approach aims to use awareness and communication of robot capabilities in order to learn a more robust approach. The paper proposes to use a graph network where each node is a robot and each edge is a communication link. The robot's observation are augmented with a vector of its capabilities (payload, sensor, etc). The method is evaluated in sim and real robotarium task where the robots must form a dense sensor network, taking into account each robot's sensor radius. The results show that the graph network and capability communication help the policy generalise to new robot configurations

**Summary Of Recommendation:**

This is an interesting paper that investigates how enabling robots to use and communicate their capabilities enables generalization to novel robot teams. The evaluation is on a fairly limited set of environments, but the results pan out.

---

### Author Response · Authors · 2023-08-15
**Common response to all reviewers and the meta reviewer**

We thank all the reviewers for their thorough and thoughtful review. Their comments have helped us make better arguments and further strengthen our work. We also thank the meta reviewer for finding appropriate reviewers and helping make final decisions.

It appears that all the reviewers generally acknowledge and appreciate the core contributions of our work, the clarity of the paper, the thoroughness of the experiments and results, and the importance of our problem of interest.

The reviewers also provided valuable suggestions and raised important questions. We will respond to each reviewer separately to address their individual questions. Below, we first address important shared comments.


**Novelty w.r.t. State Augmentation/Robot Identity (Reviewers foRA, jEiU)**

In this work, we do not attempt to make algorithmic contributions to the field of multi-agent reinforcement learning (MARL). Rather, we are concerned with uncovering insights regarding how MARL can be applied to heterogeneous multi-robot systems. Specifically, we focus on the question “How can heterogenous robots learn to coordinate in a way that enables adaptive teaming (i.e., generalize to new teams and robot types)?”. This question is highly relevant to multi-robot systems since multi-robot systems can be highly heterogeneous and might frequently involve changes to the team due to hardware failures and maintenance. The novelty of work does not solely reside in the fact that the algorithm we use augments additional relevant information as a conditioning variable. Instead, it is that we investigate the specific role of capability awareness and communication on the problem of adaptive teaming.  Importantly, our results unambiguously reveal that importance and need for reasoning about capabilities (as opposed to IDs).

It is well known in the learning community that conditioning policies and value functions on goal/task/agent identifiers improves efficiency and generalization capabilities of learned policies. For example, works in the RL community such as Bi-linear value networks [3] and Universal Value Function Approximators [2] provide deep insights into learning policies by conditioning them on aspects of the environment. However, these insights are not yet widely used within the robotics community [1]. Capabilities are inherent to all robotic systems and can be heterogeneous regardless of physical embodiment, due to differences in sensing, actuation, manufacturing, damage, or general wear-and-tear. So, we maintain that capabilities differences can include minor specification differences or major hardware and behavioral differences. It is important that we do not disregard this information, like much of the existing literature does. To our knowledge, there have been no works in the multi-robot space that specifically examine the impact of awareness and communication of capabilities as we do in this paper.

[1] M. Bettini, A. Shankar, and A. Prorok. Heterogeneous multi-robot reinforcement learning, 2023.

[2] T. Schaul, D. Horgan, K. Gregor, and D. Silver. Universal value function approximators. In Proceedings of the 32nd International Conference on Machine Learning, volume 37 of Proceedings of Machine Learning Research, pages 1312–1320, Lille, France, 07–09 Jul 2015. PMLR

[3] Z.W. Hong, G. Yang, and P. Agrawal. Bilinear value networks, 2023


**Results on new task with dependence on multiple capabilities (Reviewers TgTL, foRa, jEiU, FZZg) and more robots (Reviewer jEiU)**

We recognize that the results from the original submission could be strengthened by considering more than one task and capability. Based on reviewers’ suggestions, we carried out a full set of additional experiments on a new task which simultaneously involves a vector of multiple capabilities. Specifically, we designed and carried out an experiment on a Material Transport task, which requires specific amounts of two types of materials (lumber and concrete) to be transported between a source depot and a destination depot. The robots are rewarded for achieving the material quota without over delivering. The robots’ capabilities are defined based on the amount of lumber and concrete they can carry. The new results once conclusively demonstrate that reasoning about capabilities consistently results in better coordination and generalization than reasoning about agent IDs.

We also conducted additional experiments involving more robots as suggested. Specifically, we found that similar trends and the relative benefits of capability-awareness prevail as we increase the number of robots to 8, 10, and 15 robots.

We are unable to attach a PDF to this official top-level comment. Please see PDF attached to the official rebuttals in response to each reviewer. We will include these results in the final paper.

---

### Decision · Program_Chairs · 2023-08-30

**Decision:**

Accept (Poster)

**Comment:**

Based on the original submission, the rebuttal by the authors and the discussion that followed, the reviewers agree that this paper should be accepted to CoRL.
To the authors: great job on writing a strong rebuttal and engaging with the reviewers during the discussion period! Please address the remaining comments in the camera-ready version of the paper.